# Highly Virulent and Multidrug-Resistant *Escherichia coli* Sequence Type 58 from a Sausage in Germany

**DOI:** 10.3390/antibiotics11081006

**Published:** 2022-07-26

**Authors:** Elias Eger, Marielle Domke, Stefan E. Heiden, Madeleine Paditz, Veronika Balau, Christiane Huxdorff, Dirk Zimmermann, Timo Homeier-Bachmann, Katharina Schaufler

**Affiliations:** 1Institute of Infection Medicine, Christian-Albrecht University Kiel and University Medical Center Schleswig-Holstein, 24105 Kiel, Germany; eger@infmed.uni-kiel.de; 2Pharmaceutical Microbiology, Institute of Pharmacy, University of Greifswald, 17489 Greifswald, Germany; marielledomke@web.de (M.D.); stefan.heiden@uni-greifswald.de (S.E.H.); madeleine.paditz@uni-greifswald.de (M.P.); 3Institute of Medical Diagnostics, IMD Laboratory Greifswald, 17493 Greifswald, Germany; veronika.balau@imd-greifswald.de; 4Greenpeace e.V., 20547 Hamburg, Germany; christiane.huxdorff@greenpeace.org (C.H.); dirk.zimmermann@greenpeace.org (D.Z.); 5Friedrich Loeffler-Institut, Federal Research Institute for Animal Health, Institute of Epidemiology, 17493 Greifswald, Germany; timo.homeier@fli.de

**Keywords:** antimicrobial resistance, CTX-M-1, *Enterobacterales*, *Escherichia* *coli*, food safety, IncI1, One Health

## Abstract

Studies have previously described the occurrence of multidrug-resistant (MDR) *Escherichia* *coli* in human and veterinary medical settings, livestock, and, to a lesser extent, in the environment and food. While they mostly analyzed foodborne *E.* *coli* regarding phenotypic and sometimes genotypic antibiotic resistance and basic phylogenetic classification, we have limited understanding of the in vitro and in vivo virulence characteristics and global phylogenetic contexts of these bacteria. Here, we investigated in-depth an *E. coli* strain (PBIO3502) isolated from a pork sausage in Germany in 2021. Whole-genome sequence analysis revealed sequence type (ST)58, which has an internationally emerging high-risk clonal lineage. In addition to its MDR phenotype that mostly matched the genotype, PBIO3502 demonstrated pronounced virulence features, including in vitro biofilm formation, siderophore secretion, serum resilience, and in vivo mortality in *Galleria* *mellonella* larvae. Along with the genomic analysis indicating close phylogenetic relatedness of our strain with publicly available, clinically relevant representatives of the same ST, these results suggest the zoonotic and pathogenic character of PBIO3502 with the potential to cause infection in humans and animals. Additionally, our study highlights the necessity of the One Health approach while integrating human, animal, and environmental health, as well as the role of meat products and food chains in the putative transmission of MDR pathogens.

## 1. Introduction

Multidrug-resistant (MDR) *Enterobacterales*, such as *Escherichia coli*, which are non-susceptible to more than three classes of antibiotics [1], have been on the rise for years. They are not only found in human and veterinary clinics but also in livestock and related animal products. Therefore, they are an important component of the One Health approach [2,3], which aims at increasing and/or preserving the well-being of human, animal, and environmental health. While most One Health reports in this research area focus on the distribution of antibiotic resistance and phylogenetic backgrounds, such as sequence types (STs), little is known about phenotypic virulence traits in vitro and in vivo [4]. The latter is particularly important when considering the zoonotic nature of MDR *E. coli* from animal sources, which have the potential to infect humans, and vice versa. These pathogenic *E. coli* are divided into extraintestinal pathogenic *E. coli* (ExPEC) and intestinal pathogenic representatives (InPEC) [5], with the latter exhibiting obligate pathogenicity, and foodborne isolates potentially causing outbreaks, such as in Germany in 2011 [6,7]. In contrast, ExPEC strains usually colonize the intestinal tract of humans and animals asymptomatically but frequently cause infection outside the gut. ExPEC possess several virulence factors (VFs) that enable adhesion, biofilm formation, resilience and, thus, infection in human and/or animal hosts, and are critical for bacterial pathogenicity. However, the exact contribution of foodborne MDR ExPEC to human and animal infection and colonization has not yet been established beyond doubt, as the differentiation between commensal *E. coli* and facultative ExPEC strains remains difficult when performing either geno- or phenotypic characterization alone [8].

Here, we report an extended-spectrum β-lactamase (ESBL)-producing *E. coli* isolated from a pork sausage in Germany that we investigated in-depth by whole-genome sequencing (WGS) and functional analyses. The aim of this study was to (i) identify the phylogenetic background of the food-derived strain and explore its phylogenetic relationship to other genomes of the same sequence type, and (ii) examine the repertoire of genotypic and phenotypic resistance- and virulence-associated features combining bioinformatics with multiple in vitro and in vivo assays.

## 2. Results

### 2.1. Genomic Characterization

As part of a small screening study for which we analyzed various meat products (including fresh pork sausage) from several grocery stores in Germany in May 2021, we isolated the ESBL-producing *E. coli* strain PBIO3502, which was first verified by MALDI-TOF MS and subsequently analyzed in detail by WGS. WGS revealed that PBIO3502 belonged to ST58 (serotype O65:H16) and, thus, phylogroup B1, which includes mostly commensal but also some ExPEC and InPEC strains [9]. PBIO3502 carried plasmids with the incompatibly groups (Inc)FIA and IncI1, as well as Col plasmids. Plasmid multilocus sequence typing [10] revealed that the IncI1 plasmid was ST3 (clonal complex 3).

The presence of resistance genes was investigated using AMRFinderPlus [11] and revealed the presence of the ESBL gene *bla*_CTX-M-1_ (Ambler class A) and a BlaEC family class C β-lactamase *bla*_EC_ (98.67% identity to EC-15) in addition to genes conferring resistance to macrolides (*mdf(A)*, *mef(C)*, and *mph(G)*), streptomycin (*aadA5*), sulfonamides (*sul2*), and diaminopyrimidines (*dfrA17*). Our phenotypic analysis of antibiotic resistance was mostly consistent with the genotype (Table 1) and showed that PBIO3502 had an MDR phenotype [1]. Additionally, we noticed the presence of heavy metal resistance-associated genes (*terWZABCDE*) that enable resistance to tellurite. Usually, *E. coli* strains exhibit minimum inhibitory concentrations of tellurite of 1 µg mL^−1^ [12], but PBIO3502 showed a 64-fold increased tolerance.

Among the nearly 300 genes associated with *E. coli* virulence provided by the virulence factor database (VFDB) [13], PBIO3502 harbored several gene clusters mainly associated with adherence (CFA/I fimbriae, *cfaABCD/E*; *E. coli* common pilus, *ecpABCDER*; factor adherence *E. coli*, *fdeC*; Hsp60, *htpB*; immunogenic lipoprotein A, *ilpA*; type I fimbriae, *fimABCDEFGHIZ*; type IV pili, *vfr*), biofilm formation (alginate, *algU*; autoinducer-2, *luxS*; curli fibers, *csgABCDEFG*), invasion (*ibeBC*, *ompA*, *traJ*), iron uptake (aerobactin, *iucABCD*, *iutA*; enterobactin, *entABCDEFS*, *fepABCDEG*, *fes*; salmochelin, *iroB*), and motility (peritrichous flagella, *cheBRWYZ*, *flgBCDFGHI*, *flhABCD*, *fliAGIMPQR*, *motA*; polar flagella, *flmH*, *nueA*). Genes encoding Shiga toxin were not present.

### 2.2. Phenotypic Assessment of Virulence-Associated Traits

Given the potential zoonotic background of PBIO3502 and this large genomic set of VFs, we further characterized PBIO3502 in several phenotypic virulence-associated experiments (Figure 1).

Some bacterial populations are able to protect themselves from external stressors (e.g., UV radiation and antibiotic therapy) by forming a biofilm [14]. In addition, there is evidence that biofilm formation is a critical factor in infection development [15]. Therefore, we investigated the ability of PBIO3502 to form biofilms and analyzed whether the strain formed curli fibers (bacterial amyloid fibers) and/or cellulose (polysaccharide structures), which are important biofilm- and, thus, virulence-associated features [16]. When compared to the strong biofilm producer *E. coli* W3110 (specific biofilm formation (SBF) median, 1.96), PBIO3502 showed high adhesiveness to the plastic surface of a 96-well plate (SBF median, 2.86; Figure 1a), which likely promotes biofilm formation on catheters [17], for example. In addition, PBIO3502 showed both curli and cellulose formation in the long-term colony. The dye was used to assess the morphology of long-term colony stains only the curli fibers, but not the cellulose structures [18]. The latter are interpreted visually based on the structured surface (Figure 1b). Calcofluor, which has a high affinity for cellulose [19], was used to obtain a (semi-) quantitative result on the cellulose formation of PBIO3502. Our strain showed pronounced cellulose formation indicated by high fluorescence intensities (Figure 1c).

Iron is an important element for pathogenic bacteria, especially during infection, but also for general growth and fitness [20]. Since bacteria cannot produce iron themselves, they have to acquire it from their surroundings. Therefore, pathogens produce low-molecular iron-binding molecules (so-called siderophores) that chelate free extracellular iron(III) and are subsequently internalized by membrane receptors. We examined whether and to what extent PBIO3502 produced siderophores, and as shown in Figure 1d, PBIO3502 secreted large amounts of siderophores (mean, 87.25%). The extent of siderophores was 11.5-fold higher when compared to the negative control W3110.

Highly virulent *E. coli* representatives are capable of causing bacteremia [21] and, therefore, must compete with the host defense system (e.g., the complement system). Gram-negative bacteria use several different mechanisms to deal with the immune response provided by various mechanisms, such as capsule expression [22], down-regulation of envelope stress response pathways [23], and outer membrane surface modifications [24]. Therefore, we challenged this foodborne strain by incubating it in 50% human serum for 4 h. PBIO3502 was not inhibited by the complement-containing serum and was able to double its cell number (Figure 1e). This result again highlights the highly virulent and clinically relevant nature of PBIO3502.

In addition, we performed an infection model using larvae of the greater wax moth *Galleria mellonella* to investigate the ability of PBIO3502 to cause in vivo mortality. We injected 10^4^ or 10^5^ colony forming units (CFU) per larva. When 10^4^ CFU were injected, a mortality rate of 6.7% was obtained after 24 h (Figure 1f). This mortality rate increased to 26.7% 72 h after infection. The 10-fold increase in inoculation size resulted in an increase in the mortality rate to 36.7% at 24 h and 80.0% at 72 h of infection (Figure 1g). These results were comparable to the mortality rates of a hypervirulent *Klebsiella pneumoniae* ST420 strain PBIO2030 isolated from a wound [25]. In conclusion, the findings demonstrate the virulent potential of PBIO3502.

### 2.3. Phylogenetics of ST58

Finally, to classify our strain in a more global perspective, and again evaluate the zoonotic potential, we performed a phylogenetic comparison of an overall 159 closely related, publicly available ST58 genomes and PBIO3502 (Figure 2).

The phylogenetic tree shows that the genomes grouped into multiple clades with several subclades. Dispersed across the clades, genomes originated from humans, livestock, companion and wild animals, food, and the environment. Note that PBIO3502 clustered with three genomes from livestock (ERR3393207, ERR7420888) and environmental sources (SRR13848069), respectively. Interestingly, these closely related strains were sampled from different European locations in different years and were mainly associated with pigs. For example, ERR7420888 (designated MSG45-C09) was isolated from a healthy pig on a commercial pig farm in the Midlands of the United Kingdom in 2015 [26], while SRR13848069 (designated 20-MO00084-0) originated from scalding water from a pig slaughterhouse in Germany in 2020 [27]. All genomes assigned to the “PBIO3502 and ERR1055859 subclade” encoded aerobactin and tellurite resistance genes, a combination that was unique in the phylogenetic tree. Although PBIO3502 did not show immediate phylogenetic relatedness to human-derived strains and was in a clade primarily composed of strains from domestic and wild animals and the environment, our analysis demonstrates that phylogenetically related ST58 strains from humans were present in other clades. This again indicates the zoonotic character of the ST58 lineage and highlights the complex connections within the One Health continuum.

## 3. Discussion

A growing body of studies shows the global spread of MDR *E. coli* and underlines the clinical relevance of this pathogen, with nearly 60,000 deaths caused by antibiotic-resistant *E. coli* in 2019 alone [28]. In addition, MDR *E. coli* strains are commonly found in the environment, in animals (pets and livestock), and in meat products [2,29,30,31,32,33,34]. Due to increasing worldwide meat consumption and pressure on the prices of processed foods, slaughterhouses have to operate in an economically-oriented high throughput, potentially leading to the contamination of meat products and the water used with enteric pathogens of animal origin [35]. Therefore, it is not surprising that we and others have recently detected MDR *E. coli* strains in the wastewater of German slaughterhouses that are even resistant to the last-resort drug colistin [27,36]. Moreover, contaminated meat products might foster the transmission of pathogens from animals to humans through the food chain, of which a confirmed MDR ExPEC outbreak in London (United Kingdom) in the mid-1980s [37,38,39] and the above-mentioned outbreak in Germany in 2011 are good examples. This underlines the interdependence of animal and human health and, thus, the need for the One Health approach. Although direct consumption of unprocessed fresh pork and poultry products is uncommon in Germany, putative foodborne ExPEC transmission involves the ingestion of inadequately cooked meat or cross-contamination in the kitchen during food handling, potentially, for example, resulting in wound infection [40,41].

The establishment and frequent application of next-generation sequencing technologies have demonstrated that the spread of MDR *E. coli* appears to be driven by a limited number of pandemic STs such as ST131, ST648, or ST410 [42,43,44]. These pathogenic clonal lineages successfully combine multidrug resistance and high virulence. However, other successful clonal ExPEC lineages, such as ST58 (which belongs to clonal complex 87 [45]), increasingly emerge as globally disseminated pathogens. PBIO3502 not only exhibited numerous virulence-associated characteristics, such as strong biofilm formation, extensive siderophore secretion, and serum resistance as well as in vivo mortality, but also demonstrated a MDR phenotype, supporting the above-mentioned previous findings. Additionally note that the phenotypic results were largely consistent with predictions from our genotypic analysis.

Unlike most pandemic ExPEC lineages, which are mostly associated with phylogroups B2 or D, ST58 belongs to phylogroup B1. Interestingly, this phylogroup contains only a small number of phylogroup-specific core genes compared to other phylogroups, indicating high diversity within this phylogroup [46]. Here, we describe a foodborne *E. coli* ST58 strain that exhibits pronounced in vitro and in vivo virulence properties required for pathogenicity and bacterial survival, indicating the great potential to actually cause infection in humans and animals upon successful transmission. While the direct transfer of *E. coli* ST58 strains via the food chain has not yet been confirmed beyond doubt [4,47], and is also not within the scope of this study, it has been previously suggested that ST58 shows high clonality among strains originating from humans, animals, and the environment [48,49,50]. Our phylogenetic analysis comparing global ST58 genomes supports the latter and reiterates the clinical and zoonotic relevance of our strain. However, we used a limited number of genomes in this study. Future investigations will need to address this further and provide a more comprehensive phylogenetic and epidemiologic picture of this (putatively) successful, emerging clonal lineage.

Many resistance genes, but especially ESBL enzymes, are often encoded on plasmids, which allows rapid resistance transfer among different bacteria and, thus, widespread dissemination [51]. Interestingly, PBIO3502 carried an IncI1 plasmid, which appears to be a common plasmid type of InPEC strains, but also, to a lesser extent, occurs in ExPEC [52]. Moreover, ST3 IncI1 plasmids are not only frequently found in strains isolated from humans but also in pathogens of animal origin [53]. This is further evidence of the zoonotic character of PBIO3502.

## 4. Materials and Methods

### 4.1. Strain Origin and General Methods

As part of a small screening study, packaged fresh meat samples from various grocery stores were tested for different MDR bacteria. Briefly, the meat samples were opened under sterile conditions, two pieces of approximately 5 mm³ were cut out with a sterile scalpel (Braun, Melsungen, Germany) and transferred to 5 mL of tryptic soy broth (Carl Roth, Karlsruhe, Germany). The bacteria were then enriched under shaking conditions (130 rpm) at 37 °C overnight. Then, 100 µL of the bacterial suspension was plated onto different chromogenic selection plates and incubated overnight at 37 °C. The strain 27-ESBL-EC (internal designation PBIO3502) was cultivated on a CHROM-ESBL selection plate (Mast Diagnostica GmbH, Reinfeld, Germany) and the species *E. coli* was confirmed using MALDI-TOF MS (Bruker, Bremen, Germany). It was derived from a fresh bratwurst purchased from a German grocery store. All strains were stored at −80 °C in lysogeny broth (LB; Carl Roth, Karlsruhe, Germany) supplemented with 20% (*v*/*v*) glycerol (anhydrous; Merck, Darmstadt, Germany). Prior to use, one single colony of fresh overnight cultures on LB agar plates was inoculated in 5 mL of LB and grown under shaking conditions (130 rpm) at 37 °C overnight.

### 4.2. Whole-Genome Sequencing

Total DNA was extracted using the MasterPure DNA Purification Kit for Blood, v. 2 (Lucigen, Middleton, WI, USA), according to the manufacturer’s instructions. The isolated DNA was quantified fluorometrically using the Qubit 4 fluorometer and the corresponding dsDNA HS Assay Kit (Thermo Fisher Scientific, Waltham, MA, USA). DNA was shipped to the Microbial Genome Sequencing Center (MiGS), now SeqCenter (Pittsburgh, PA, USA), and after library preparation using the Illumina DNA Prep Kit and IDT 10 bp UDI indices (Illumina, San Diego, CA, USA) sequenced on an Illumina NextSeq 2000, producing 2 × 151 bp reads. Demultiplexing, quality control, and adapter trimming were performed using bcl-convert v. 3.9.3 [54].

### 4.3. Sequence Assembly and Genomic Analyses

Short-read data were processed using BBDuk from BBTools v. 38.95 (https://sourceforge.net/projects/bbmap/, accessed on 3 March 2022) to trim the adapters, filter possible PhiX contaminants, and do further quality- and polymer-trimming. Quality control (QC) of provided (raw) and processed (trimmed) reads was performed using FastQC v. 0.11.9 (https://www.bioinformatics.babraham.ac.uk/projects/fastqc/, accessed on 3 March 2022). Trimmed reads were assembled using shovill v. 1.1.0 (https://github.com/tseemann/shovill, accessed on 3 March 2022) with SPAdes v. 3.15.3 [55]. An additional polishing step (besides the one implemented in the shovill assembly pipeline) was performed by first mapping the trimmed reads to the assembly using BWA v. 0.7.17 [56]. The alignment files were then converted to binary format, sorted, and duplicate reads were marked with SAMtools v. 1.14 [57]. Finally, the draft contigs were corrected using Pilon v. 1.24 [58]. Genome completeness and contamination were assessed using CheckM v. 1.1.3 [59]. Prokka v. 1.14.6 [60] was used to automatically annotate the draft assembly. Genomic analyses including in silico multilocus sequence typing, serotype prediction, and antibiotic resistance and virulence feature detection were performed using mlst v. 2.19.0 (https://github.com/tseemann/mlst, accessed on 3 March 2022; with the PubMLST [61] database and Enterobase [62]), ABRicate v. 1.0.0 (https://github.com/tseemann/abricate, accessed on 3 March 2022; with ResFinder [63], PlasmidFinder [10], VFDB [13], BacMet2 [64], and EcOH [65] databases), and AMRFinderPlus v. 3.10.30 with database v. 2022-05-26.1 [11]. Assignment to the phylogroup was performed using ClermonTyping v. 20.03 [66].

### 4.4. Phylogeny

For the creation of a core single-nucleotide polymorphism (SNP)-based phylogeny of closely related isolates, Enterobase [62] was searched for *E. coli* ST58 strains (*n* = 2199), for which paired Illumina read data were available (*n* = 2186). Data were downloaded from the public European Nucleotide Archive SRA FTP server (ftp://ftp.sra.ebi.ac.uk/vol1/, accessed on 20 June 2022). Mash v. 2.3 [67] was used to generate sketches (k-mer size, 21; sketch size, 1,000,000; minimum number of k-mers for reads, 3) of the reference sequence (draft genome of PBIO3502) and the public raw read data. Afterwards, the distance between the reference sketch and the query raw read sketches was estimated and the 200 read sets with the shortest distance (most shared k-mers) selected. For these accessions, read trimming, genome assembly, and analyses were performed as described above. The trimmed reads were mapped against the draft genome of PBIO3502 using snippy v. 4.6.0 (https://github.com/tseemann/snippy, accessed on 22 June 2022) to create a whole-genome alignment with 201 sequences. The alignment was processed using Gubbins v. 3.2.1 [68] to filter out regions with SNPs that are likely the result of recombination. In this step, two isolates were removed due to alignment gaps or missing information at more than 25% of the alignment sites. The alignment of the remaining sequences (*n* = 199) was processed using snp-sites v. 2.5.1 [69] to retain only alignment positions containing A, C, G, or T exclusively. A maximum likelihood tree was inferred with RAxML-NG v. 1.1.0 [70] using GTR+G by first parsing the alignment and excluding 40 sequences that were completely identical to another sequence. The final alignment (containing 159 sequences and 4984 sites) was then processed by searching 500 parsimony and 500 random starting trees and performing 1000 bootstrap repeats. The best-scoring maximum likelihood tree was midpoint-rooted in iTOL v. 6.5.7 [71] and visualized with bootstrap support values and metadata. The exported graphic was post-processed using Affinity Designer v. 1.10.5.1342 (https://affinity.serif.com/en-us/designer/, accessed on 22 June 2022).

### 4.5. Minimum Inhibitory Concentration

Phenotypic antimicrobial susceptibility testing was performed in collaboration with the IMD Laboratory Greifswald (Germany) using the automated VITEK 2 system (bioMérieux, Marcy l’Etoile, France). Testing was performed using the AST-N389 card, according to the manufacturer’s instructions. MIC values of gentamicin (Carl Roth, Karlsruhe, Germany), streptomycin (Sigma-Aldrich, St. Louis, MO, USA), tetracycline (Carl Roth, Karlsruhe, Germany), and tellurite (Alfa Aesar, Haverhill, MA, USA) were determined by broth microdilution according to ISO Standard 20776-1 [72]. Briefly, several single colonies were resuspended in a 0.9% (*w*/*v*) NaCl solution until the corresponding suspensions had an OD_600_ equal to 0.5 McFarland standard turbidity. The bacterial suspensions were then diluted 1:230 in cation-adjusted Mueller–Hinton broth 2 (MH-2; Sigma-Aldrich, St. Louis, MO, USA), corresponding to approximately 10^5^ CFU mL^−1^. Serial 2-fold dilutions were performed in a 96-well plate (Sarstedt, Nümbrecht, Germany) with concentrations ranging from 64 to 0.5 µg mL^−1^ for the antibiotics gentamicin, streptomycin, and tetracycline, and from 512 to 0.5 µg mL^−1^ for the heavy metal tellurite, respectively. Finally, the bacterial suspensions were added and incubated at 37 °C for 18 ± 2 h. The MIC represents the lowest concentration of antimicrobial agent that inhibits visible bacterial growth.

### 4.6. Biofilm Formation

Biofilm formation on polystyrene surfaces was determined by crystal violet (CV) staining, as previously described [73,74] with some adjustments. First, overnight cultures were diluted 100-fold in 5 mL of fresh M9 minimal salt medium (MP Biomedicals, Irvine, CA, USA) containing 0.4% (*w*/*v*) glucose (Carl Roth, Karlsruhe, Germany) and incubated at 37 °C and 130 rpm until OD_600_ reached 0.5 McFarland standard turbidity. The bacterial suspensions were then diluted 1:10 and 200 µL were then transferred in triplicate to a 96-well flat-bottomed polystyrene plate (Nunc, Thermo Fisher Scientific, Waltham, MA, USA). In addition, three control wells were filled with 200 µL of sterile medium. The plates were hermetically sealed and incubated at 28 °C for 24 h without shaking. Subsequently, OD_600_ was measured using CLARIOstar Plus (BMG LABTECH GmbH, Ortenberg, Germany), planktonic cells were removed by washing three times with deionized water, and the microtiter plates were air-dried for 10 min. After fixation with 250 µL of 99% (*v*/*v*) methanol (Merck, Darmstadt, Germany) for 15 min and air-drying, the cells were stained with 250 µL of a 0.1% (*w*/*v*) aqueous CV solution (Sigma-Aldrich, St. Louis, MO, USA) for 30 min. The staining solution was then discarded, and unbound dye was removed by washing three times with deionized water. After air-drying for 10 min, the bound CV was dissolved with a mixture of 80 parts ethanol (99.8% (*v*/*v*); Carl Roth, Karlsruhe, Germany) and 20 parts acetone (Merck, Darmstadt, Germany) at room temperature (rt) with horizontal shaking at 200 rpm for 30 min. After complete dissolution, 125 µL were transferred to a new microtiter plate and optical density was measured at a wavelength of 570 nm using the plate reader. The strength of biofilm formation was expressed as specific biofilm formation (SBF). The SBF was calculated according to the following formula [75]: SBF = (B−NC)/G, where B is the OD_570_ of the stained bacteria, NC is the OD_570_ of the stained control wells to eliminate the fraction of CV adhering to the polystyrene surface due to abiotic factors, and G is the OD_600_ representing the density of cells grown in the media.

The expression of biofilm-associated extracellular matrix components such as cellulose and curli fibers was tested using macrocolony assays as previously described [73]. Five microliters of overnight culture was dropped onto span agar plates (Hellmuth Carroux, Hamburg, Germany). The plates were hermetically sealed and incubated at 28 °C for 5 days. For detection of curli fimbriae, 0.005% Congo red and 0.0025% Coomassie Brilliant Blue G-250 (both chemicals were purchased from Carl Roth, Karlsruhe, Germany) were added to the span agar and evaluated visually. For cellulose staining, the span agar contained 0.004% Calcofluor (Sigma-Aldrich, St. Louis, MO, USA), and the extent of cellulose formation was measured by fluorescence intensity using the plate reader (excitation, 400–415 nm, emission, 480–520 nm).

### 4.7. Siderophore Production Assay

Quantification of the extent of siderophores secreted by PBIO3502 was performed according to a previously published protocol [23]. Briefly, the adjusted bacterial suspensions were diluted and grown in 5 mL of chelated M9 minimal medium supplemented with casamino acids (c-M9-CA) at 37 °C and 130 rpm for 24 h. The c-M9-CA consisted of the following compounds: M9 minimal salt medium (MP Biomedicals, Irvine, CA, USA), 2 mM MgSO_4_ (Carl Roth, Karlsruhe, Germany), 200 µM 2,2′-dipyridyl (Carl Roth, Karlsruhe, Germany), and 0.3% (*w*/*v*) casamino acids (BD, Franklin Lakes, NJ, USA). After incubation, 1 mL of the bacterial cultures were centrifuged (4900× *g* for 20 min at rt) and 100 µL of the siderophore-containing supernatant were transferred in triplicate to 96-well microtiter plates (Nunc, Thermo Fisher Scientific, Waltham, MA, USA) with 100 µL of chrome azurol S shuttle solution (composition according to [76]) already in the wells. Wells containing only fresh medium and 15 mM EDTA (Carl Roth, Karlsruhe, Germany) served as a blank and positive control, respectively. The non-siderophore producer W3110 served as a negative control. All mixtures were incubated in the dark at rt for 30 min. Finally, absorbance was measured at a wavelength of 630 nm using the plate reader. Secretion of siderophores was calculated as previously described [77] and expressed as a percentage unit of siderophore production.

### 4.8. Serum Resistance

Determination of survival in 50% human serum was performed as previously described [23]. Briefly, overnight cultures were diluted 1:100 in 5 mL of fresh LB and incubated with shaking at 37 °C until OD_600_ reached 0.5 McFarland standard turbidity. Then, 1000 µL of the bacterial suspension was pelleted (7500× *g* for 5 min at rt) and resuspended in 1 mL of phosphate-buffered saline (PBS; Thermo Fisher Scientific, Waltham, MA, USA). One hundred microliters of the sample were seeded in a 96-well microtiter plate containing 100 µL of human serum (United States origin; Sigma-Aldrich, St. Louis, MO, USA) per well (resulting in a final concentration of 50% human serum and approximately 10^8^ CFU mL^−1^). Following this, 20 µL of each sample was withdrawn and serial dilutions were plated on LB agar plates and incubated overnight at 37 °C to determine the size of the inoculum. The inoculated microtiter plates were incubated for 4 h at 37 °C without agitation. Thereafter, the number of surviving CFU mL^−1^ was determined by plating out serial dilutions and incubating at 37 °C overnight. The positive control in each experiment was the serum-resistant PBIO1289 (initially designated as IMT10740 [78,79]). The serum-sensitive W3110 served as the negative control. Serum resistance was expressed as log_2_ fold change in CFU mL^−1^ after treatment with respect to inoculum size.

### 4.9. Infection of Galleria mellonella Larvae

Infections of larvae of the greater wax moth *G. mellonella* were performed as previously described [23]. Overnight cultures were diluted 1:100 in 30 mL of fresh LB and incubated with shaking at 37 °C to an OD_600_ of 1.0. Then, 1000 µL of the bacterial suspension were pelleted (16,000× *g* for 5 min at rt) and washed twice with PBS. The bacterial suspensions were diluted to 10^6^ CFU mL^−1^ and 10^7^ CFU mL^−1^, respectively. Larvae (proinsects, Minden, Germany) were randomly divided into groups of 10 individuals each and 10 µL of the adjusted bacterial suspensions were injected into the left proleg. In addition, 10 µL of PBS was injected into a group of larvae to ensure that death was not due to trauma from the injection. Each group was placed in 90 mm glass Petri dishes, kept at 37 °C in the dark, and death was recorded every 24 h. Individuals were considered dead when they no longer responded to physical stimuli and showed pigmentation. The results of three independent tests were pooled for each strain to generate Kaplan–Meier plots of mortality rates [80].

### 4.10. Data Visualization and Analysis

Data visualization was performed using GraphPad Prism v. 9.3.1 for macOS (GraphPad Software, San Diego, CA, USA). All experiments were performed with three or more independent biological replicates. Unless otherwise specified, data were expressed as mean and standard error.

## 5. Conclusions

The finding of a highly virulent and MDR, foodborne *E. coli* ST58, which was phylogenetically related to clinical strains of the same ST, suggests its zoonotic and pathogenic potential and the relevance of the One Health approach. Additionally, our study highlights the potential role of food (chains) in the spread and putative transmission of MDR pathogens.

## Figures and Tables

**Figure 1 antibiotics-11-01006-f001:**
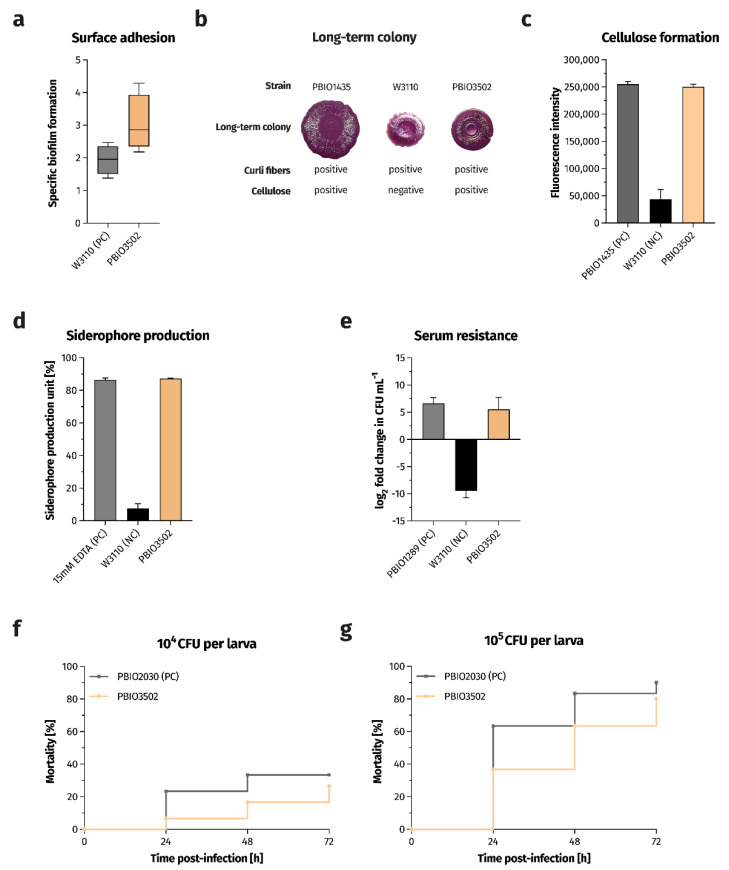
PBIO3205 exhibits a highly virulent phenotype. (**a**) Biofilm formation on polystyrene surfaces as determined by crystal violet staining (*n* = 4). Results are expressed as growth-adjusted specific biofilm formation. The line within the box marks the median value, while the boxplot represents the 25th to 75th percentile of the data set. The whiskers mark the minimum and maximum values, respectively. (**b**) Morphology of long-term colonies examining the expression of the biofilm-associated extracellular matrix components curli fibers and cellulose. Congo red was used to specifically stain curli fibers, and cellulose was visually interpreted based on the textured surface. PBIO1435 and W3110 were used as references. (**c**) Cellulose formation was examined by staining with Calcofluor and measuring the fluorescence intensity of bound Calcofluor (*n* = 3). Results are given as mean values of fluorescence intensity and standard errors. (**d**) The extent of secreted siderophores is expressed as the mean of the percent unit of siderophore production and standard error (*n* = 3–5). (**e**) Survival in 50% human serum (*n* = 3–5). Results are given as means and standard errors of log_2_ fold change in CFU mL^−1^ after 4 h of incubation in the presence of human serum. (**f**,**g**) Kaplan–Meier plot of mortality in the *Galleria mellonella* larvae infection model (*n* = 3). Results are expressed as mean percent mortality after injection of 10^4^ CFU per larva (**g**) and 10^5^ CFU per larva, respectively. NC, negative control. PC, positive control.

**Figure 2 antibiotics-11-01006-f002:**
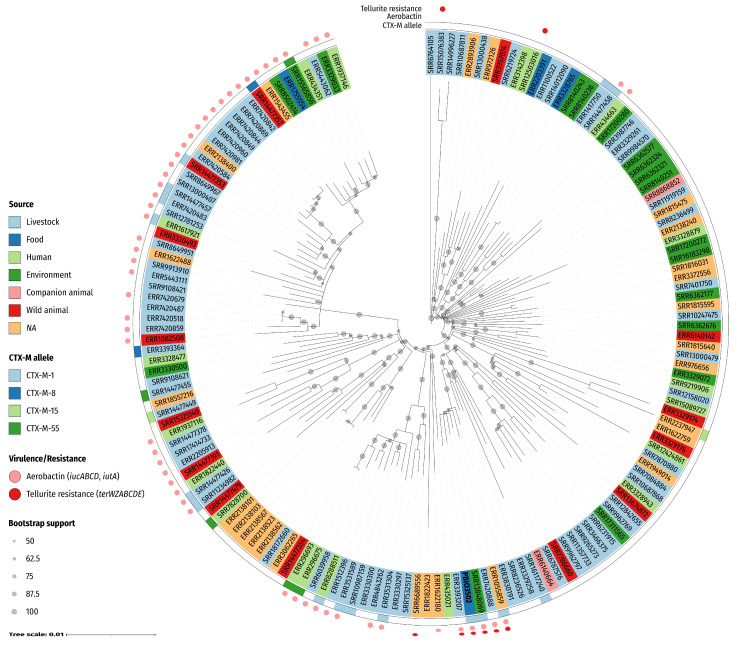
Phylogenetics reveals close relationship of PBIO3502 with publicly available genomes (*n* = 159). Included genomes were selected based on most shared k-mers using Mash. The phylogenetic tree was constructed using a maximum likelihood-based approach and is based on a core singe-nucleotide polymorphism alignment (4984 sites). The circular tree was midpoint-rooted and the circles on the branches indicate bootstrap support of ≥50% from 1000 replicates. The labels indicate the accession number (except PBIO3502) and are colored according to their source/host, as indicated in the legend. Annotations indicate (from inner to outer circle): encoded CTX-M (cefotaximase) allele, presence of genes encoding aerobactin and tellurite resistance. *NA*, not applicable (i.e., no metadata provided).

**Table 1 antibiotics-11-01006-t001:** Phenotypic and genotypic resistance profile of PBIO3502.

Antimicrobial Category	Antimicrobial Agent	MIC ^a^ [µg mL^−1^]	S/R ^b^	Genotype ^d^
Aminopenicillin + β-lactamase inhibitor	Ampicillin/sulbactam	≥32/16	R	*bla* _EC_
Ureidopenicillin + β-lactamase inhibitor	Piperacillin/tazobactam	≥128/4	R	*bla* _EC_
Third generation cephalosporins	Cefotaxime	≥64	R	*bla* _CTX-M-1_
	Ceftazidime	≥64	R	
Carbapenems	Meropenem	≤0.25	S	
Aminoglycosides	Gentamicin	≤0.5	S	
	Streptomycin	64	*NA*	*aadA5*
Fluoroquinolones	Ciprofloxacin	≤0.25	S	
Tetracyclines	Tetracycline	≤0.5	S ^c^	
Folate pathway inhibitors	Trimethoprim/sulfamethoxazole	≥16/304	R	*dfrA17*, *sul2*
Phosphonic acids	Fosfomycin	≤16	S	

^a^ MIC, minimum inhibitory concentration; ^b^ Interpretive categories according to EUCAST (The European Committee on Antimicrobial Susceptibility Testing. Breakpoint tables for interpretation of MICs and zone diameters. Version 12.0. 2022). S, susceptible; R, resistant; *NA*, not applicable. ^c^ Result was interpreted according to the published breakpoints of the CLSI (Clinical and Laboratory Standards Institute. Performance standards for antimicrobial susceptibility testing. 32nd edition. 2022). ^d^ Prediction based on alignment of sequences from AMRFinderPlus database (threshold for coverage and identity, ≥80%).

## Data Availability

The data for this study have been deposited in the European Nucleotide Archive (ENA) at EMBL-EBI under accession number PRJEB53849 (https://www.ebi.ac.uk/ena/browser/view/PRJEB53849, uploaded on 24 June 2022).

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
