# Peer review of "Highly Virulent and Multidrug-Resistant Escherichia coli Sequence Type 58 from a Sausage in Germany"

_antibiotics, 2022, doi:10.3390/antibiotics11081006_

Round 1

Reviewer 1 Report

Thank you for the opportunity to review this article. The authors aimed to investigate an E. coli strain (PBIO3502) isolated from a pork sausage in Germany and the results suggested the zoonotic and pathogenic character of PBIO3502 with the potential to cause infection in humans and animals. The quality of this manuscript is excellent, I agree to be published in the journal of Antibiotics. There has only a minor mistake in manuscript as follow:

In line 96, “invasion (ibeBC, ompA, traJ) iron uptake” should be revised to “invasion (ibeBC, ompA, traJ) , iron uptake”.  

Author Response

Point 1: In line 96, “invasion (ibeBC, ompA, traJ) iron uptake” should be revised to “invasion (ibeBC, ompA, traJ) , iron uptake”. 

Response 1: We appreciate the time and effort you have taken to provide feedback and are very grateful for your suggestion to improve our manuscript. We changed this accordingly.

Reviewer 2 Report

pdf file

Author Response

Reviewer 2: In my opinion, this Manuscript is almost ready for publication and the undertaken topic will be interesting for the Readers of Antibiotics.

Response: We appreciate the time and effort you have spent on your feedback and are very grateful for the valuable comments to improve our manuscript. We have implemented your suggestions and highlighted the appropriate changes in the manuscript. In addition, you will find below in red our point-by-point response to your comments and concerns.

Point 1: Line 22, Line 40: in vitro and in vivo please use italics

Response 1: Apologies. You are absolutely right, we adapted this accordingly.

Point 2: Line 380: Why did the Authors use 50% of human serum? I know that the protocol was adapted from previous publications, and I am not denying it. I also see in other literature that the use of 50% of human serum is a common practice. What, according to the Authors, does this result from? How does this relate to the application potential? This is my more personal question that has no impact on the quality of Your publications

Response 2: Thank you for this interesting question. The idea of testing serum resistance in our study was to challenge our isolate with the complement-containing serum for only four hours. However, the concentration we chose is much higher than under natural circumstances, but as you noted, it is consistent with previously published protocols. It has been our experience that as serum concentration decreases, the differences between isolates with numerous other virulence properties and isolates without these properties decrease significantly. In particular, when isolates are incubated in serum for only four hours, (statistical) differentiation is sometimes no longer possible. On the other hand, an extension of the incubation time (e.g., 24 hours) is not suitable, since particularly fast-growing strains may reach the late stationary phase or the death phase and the decreasing CFU is thus based on this rather than on the result of complement-mediated killing. We hope this contributes to a better understanding.

Point 3: Line 369 and Line 400: The Authors use units inconsistently once there is mL, then µl, please standardize

Response 3: Thank you, we improved this accordingly.

Reviewer 3 Report

This manuscript focused on the phenotypic and genotypic characterization of a highly virulent and multidrug-resistant Escherichia coli from a sausage. It was generally well written and should be of interest to the readers. I only offer the following suggestions:

(1) Please reduce the number of Keywords.

(2) There were some discussions in the Results section with references. I would suggest removing these descriptions to the Discussion section.

(3) Lines 61: Some subtitles should be added for the ease of readers. Moreover, it would be better to put the phylogenetics studies right after the genomic analysis of PBIO3502.

(4) Line 186: There were many parts that were not directly related to the main findings of this work. Thus, the Discussion section is required to be reorganized. Analysis of the correlation between phenotypic and genotypic data can be included in this part.

(5) Line 436: The number of References can be reduced since the article type is “Communication”.

Author Response

Reviewer 3: This manuscript focused on the phenotypic and genotypic characterization of a highly virulent and multidrug-resistant Escherichia coli from a sausage. It was generally well written and should be of interest to the readers.

Response: We appreciate the time and effort you have spent on your feedback and are very grateful for the valuable comments to improve our manuscript. We have implemented your suggestions as best as possible and highlighted the appropriate changes in the manuscript. In addition, you will find below in red our point-by-point response to your comments and concerns.

Point 1: Please reduce the number of Keywords.

Response 1: Thank you for this suggestion. The author guidelines of Antibiotics allow the use of a minimum of three and a maximum of ten keywords. Therefore, we initially decided to use the selected keywords. However, we have refined our selection and removed the keywords "extended-spectrum β-lactamase", "next-generation sequencing", and "virulence".

Point 2: There were some discussions in the Results section with references. I would suggest removing these descriptions to the Discussion section.

Response 2: Again, thank you very much for this comment. In order to give a clear narrative of our study, we decided to include some descriptions in the Results section and would like to keep them for clarity. However, we have not adequately discussed some of our findings. As suggested by you at point 4, we have included a comparison between the genotypic and phenotypic results in the Discussion section: “PBIO3502 not only exhibited numerous virulence-associated characteristics such as strong biofilm formation, extensive siderophore secretion, and serum resistance as well as in vivo mortality, but also demonstrated a MDR phenotype, supporting the above-mentioned previous findings. Also note that the phenotypic results were largely consistent with predictions from our genotypic analysis.” (lines 213–218)

Point 3: Lines 61: Some subtitles should be added for the ease of readers. Moreover, it would be better to put the phylogenetics studies right after the genomic analysis of PBIO3502.

Response 3: We now included appropriate subtitles in the Results section: "Genomic characterization" (line 62), "Phenotypic assessment of virulence-associated traits" (line 101), and "Phylogenetics of ST58" (line 160). We believe that redesigning the structure of the Results section, with the phylogenetic analysis immediately following the genomic analysis, would compromise the clarity of our study and would thus like to keep it as it is.

Point 4: Line 186: There were many parts that were not directly related to the main findings of this work. Thus, the Discussion section is required to be reorganized. Analysis of the correlation between phenotypic and genotypic data can be included in this part.

Response 4: Apologies, you are definitely right. As described before, we have now included this comparison in the Discussion section.

Point 5: Line 436: The number of References can be reduced since the article type is “Communication”.

Response 5: We appreciate the reviewer's suggestion, but we respectfully disagree. The journal's author guidelines state that a communication must contain "around 2000 words at minimum, with at least 20 references". No maximum limit is specified. We admit that we included numerous references in the manuscript. This extensive literature search is intended to support the scientific quality of our work and we would therefore be reluctant to remove references.

Reviewer 4 Report

Very interesting and well structured work.

Author Response

Reviewer 4: Very interesting and well structured work.

Response: Thank you for taking your time to assess our manuscript and provide us with your feedback.